# The Role of GNSS-RTN in Transportation Applications

Sajid Raza [1], Ahmed Al-Kaisy [2,*], Rafael Teixeira [1] and Benjamin Meyer [1]

[1] Western Transportation Institute, Montana State University, Bozeman, MT 59717, USA; sajid.raza@montana.edu (S.R.); rafaeldeoft.teixeira@student.montana.edu (R.T.); benjamin.meyer3@student.montana.edu (B.M.)

[2] Department of Civil Engineering, Montana State University, Bozeman, MT 59717, USA

* Correspondence: alkaisy@montana.edu; Tel.: +1-(406)-994-6116

**Definition:** The Global Navigation Satellite System—Real-Time Network (GNSS-RTN) is a satellite-based positioning system using a network of ground receivers (also called continuously operating reference stations (CORSs)) and a central processing center that provides highly accurate location services to the users in real-time over a broader geographic region. Such systems can provide geospatial location data with centimeter-level accuracy anywhere within the network. Geospatial location services are not only used in measuring ground distances and mapping topography; they have also become vital in many other fields such as aerospace, aviation, natural disaster management, and agriculture, to name but a few. The innovative and multi-disciplinary applications of geospatial data drive technological advancement towards precise and accurate location services available in real-time. Although GNSS-RTN technology is currently utilized in a few industries such as precision farming, construction industry, and land surveying, the implications of precise real-time location services would be far-reaching and more critical to many advanced transportation applications. The GNSS-RTN technology is promising in meeting the needs of automation in most advanced transportation applications. This article presents an overview of the GNSS-RTN technology, its current applications in transportation-related fields, and a perspective on the future use of this technology in advanced transportation applications.

**Keywords:** GNSS-RTN; real-time network; highly accurate geospatial data; transportation





## 1. Introduction

In the past few decades, significant technological advances have been made in global navigation satellite systems (GNSS), which include the Global Positioning System (GPS) (U.S. GNSS) and its counterparts: Globalnaya Navigazionnaya Sputnikovaya Sistema (GLONASS, Russia), Galileo (Europe), Quasi-Zenith Satellite System (QZSS, Japan), and BeiDou (China) [1]. The GNSS has become one of the fastest-growing emerging technologies delivering location services to various industries. Geospatial data are not only used in measuring ground distances and mapping topography [2], but they have also found significant applications in other fields such as agriculture, construction, mining, bridge health monitoring, natural disaster management [3], and accurate navigation [4]. Among all these fields, geospatial technology plays a remarkable role in the transportation sector and has the potential to play an even more critical role in future autonomous transportation systems. This article sheds light on the major existing GNSS-RTN transportation applications and provides an outlook on the future role this technology plays in advanced transportation systems. In this article, the GPS is occasionally used to refer to the GNSS technology in the broader sense and not necessarily in reference to the U.S. constellation of navigation satellites. In these instances, the use of GPS is deemed more appropriate, as it involves the use of globally accepted technical terminologies.

This article aims to compile and synthesize the literature on the use of GNSS-RTN geospatial data in transportation applications. It provides a concise high-level overview to engineers and professionals working in the transportation industry.

## 2. GNSS RTN: State of Technology

With the emergence of GPS Real-Time Kinematic (RTK) technology in the early 1990s, the use of GPS-RTK became vital in various applications which require accurate location data in real-time. However, the performance and accuracy of the traditional GPS-RTK are limited due to the distance between a reference station (a.k.a. base station) and a roving receiver (user device). The GPS-RTK is a positioning technique that uses a fixed base station placed at a known location which transmits correction to the rover to improve accuracy and minimize errors. The accuracy and reliability of the GPS-RTK measurements degrade with the increase in baseline length (i.e., base-to-rover distance) due to distance-dependent errors and biases. To achieve more reliable and accurate results, specifically, centimeter-level accuracy using the GPS-RTK technique, it is required that the roving receiver (rover) is located within a restricted range (typically in the order of 10 km) of the reference station [5]. To overcome the limitation of the baseline length of the traditional GPS-RTK technique and with the advancement in GNSS technology, the GNSS-RTN concept was introduced in the mid-1990s [6].

The GNSS-RTN is a satellite-based positioning system using a network of ground receivers (also called base stations, reference stations, or continuously operating reference stations (CORSs)) to improve the accuracy of corrections in positioning data. This concept is shown in Figure 1. The network of reference stations extenuates and alleviates the spatially-correlated atmospheric and satellite orbit biases [7] and improves the accuracy and precision of geospatial positioning through real-time corrections sent from a central processing center to a rover. The utilization of ground sensors enables systems to have a range of 1 to 5 cm in accuracy, compared to a range of 1 to 10 m when sensors are not utilized [8].

With the technological advancement in satellite systems, the use of GNSS-RTN technology for the correction of positioning data has evolved into commercially viable systems available today. The advent of GNSS-RTN systems made it possible to achieve highly accurate positioning over a distance of 50–70 km (reference station spacing should generally not exceed 70 km) from the base station [9].

Since the advent of the GNSS-RTN technology, national networks have been established in many countries around the world, especially in developed countries. In the United States, the National Oceanic and Atmospheric Administration (NOAA) CORS Network (NCN) is the cornerstone of the geometric component of the National Spatial Reference System (NSRS), with observations from over 2800 stations nationwide. These CORSs are part of subnetworks operated by 239 public and private entities [10]. In addition, major technology vendors are offering GNSS-RTN products and location-based services (LBSs), with Leica Geosystems, Trimble, and Topcon being the three most pervasive providers of GNSS-based services and products around the globe [11]. Figure 2 shows the coverage of the Leica Geosystems network available to users in the U.S.

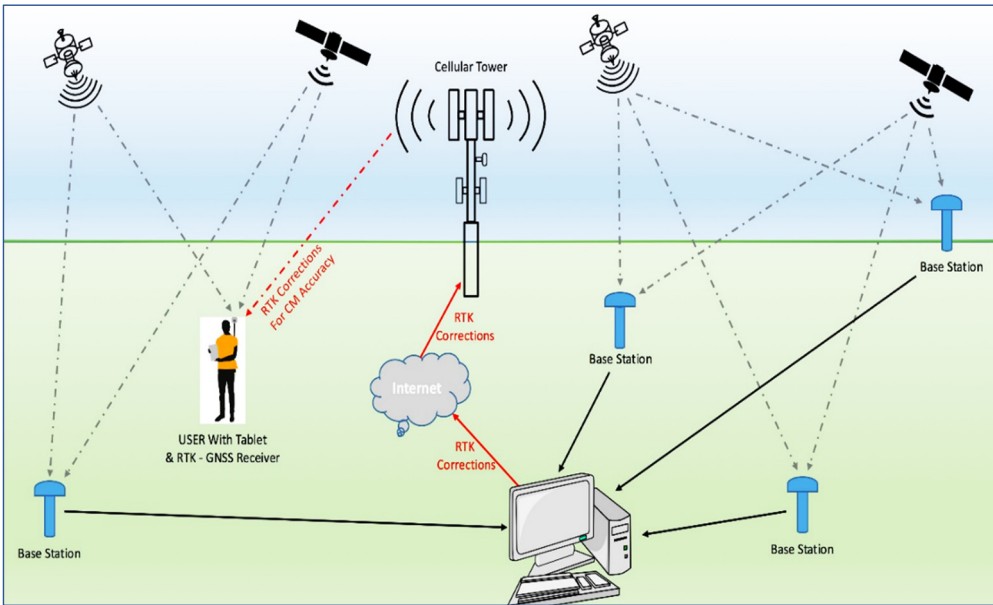

**Figure 1.** Concept of GNSS-RTN Operation, Reprinted from Anatum [12].

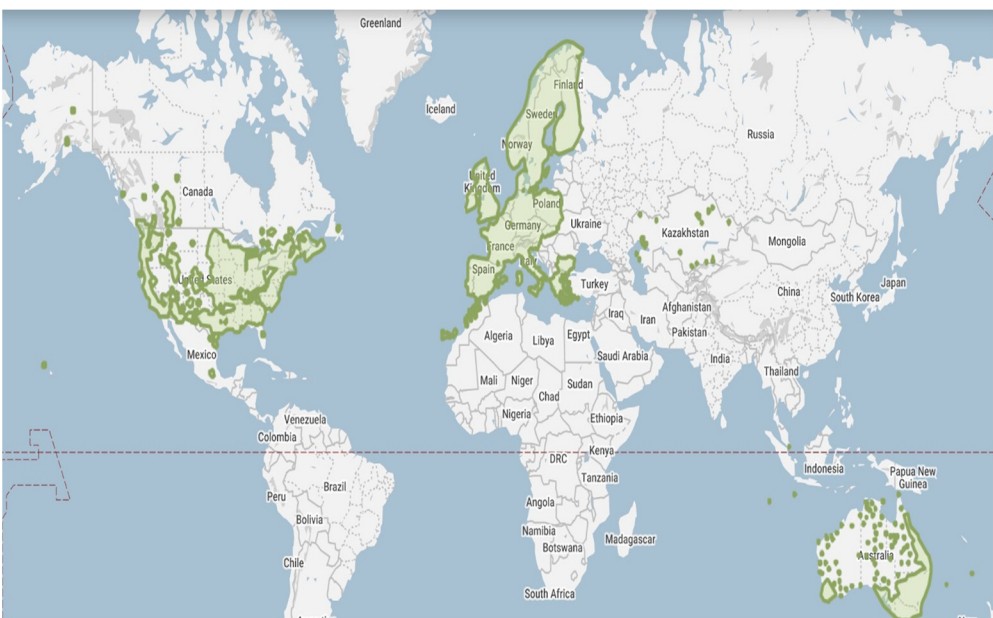

**Figure 2.** HxGN SmartNET Global Coverage, Reprinted from Leica [13].

*GNSS-RTN Versus Other Positioning Methods*

The geospatial location data obtained using the various GNSS positioning methods normally involve inaccuracies (errors) that are inherent to the GNSS system, the positioning method, and the environment. The most important sources of errors affecting GNSS signals include clock errors and signal propagation errors. Examples of signal propagation errors include ionospheric errors (errors caused by the propagation time delay when the signal passes through the ionosphere), tropospheric errors (errors caused by the signal passing through the dry gases and water vapor of the troposphere), and multipath errors (errors caused when the signal reaches the receiver's antenna via more than one path). In addition, system errors, intentional error sources such as selective availability, signal jamming, and signal spoofing affect the accuracy of positioning data. While not a source of error, the dilution of precision (DOP) factor (a factor related to the geometry of visible satellites) can also affect the accuracy of the final results [14].

Various types of geospatial correction services are available to address GNSS users' requirements for increased accuracy, signal availability and system integrity. Standalone GNSS is the standard GNSS practice, also known as Single Point Positioning (commonly referred to as GPS). It has no error corrections, and GNSS satellites just provide the best standard signals available to the rover; therefore, it cannot be used indoors or in urban environments where there are many signal obstructions. The GPS's integrity cannot be assured [15]. Hence, for critical user applications such as air and marine navigation, GPS-only positioning is not recommended and must be augmented in some way. The accuracy of standalone GNSS is up to several meters.

To increase the accuracy of location data, the RTK or differential GPS (DGPS) is used to provide positioning corrections to standalone GNSS signals. The RTK positioning method offers 1–2 cm accuracy [16]. However, the receiver needs to work within a short range from the base station. In addition, no redundancy of the base station is usually available if the base station experiences any malfunctioning [16]. Another GNSS positioning technique that is accessible worldwide is called precise point positioning (PPP). The PPP removes GNSS system errors based on GNSS satellite clock and orbit corrections, which are generated from a network of reference stations around the globe. The corrections are delivered to the end-user via satellite or through the Internet, resulting in cm-level accuracy with no local or regional ground base station infrastructure requirements. However, the initialization time could take up to 30 min, which makes PPP impractical for certain applications that require much faster positioning data, such as surveying and construction. Some PPP correction services only provide corrections for satellite clock and orbit errors, which lowers the accuracy of measurements. A hybrid method that has also been used combines the PPP's global access with the accuracy and quick initialization times of near-RTK. It relies on a network of reference stations within about 150 km of each other (i.e., similar to GNSS-RTN CORSs but with a lower density network). The CORSs collect GNSS data and calculate both satellite and atmospheric correction models.

The last GNSS positioning method discussed here is the GNSS-RTN system, which offers high accuracy and minimum initialization time over a broader coverage area. The GNSS_RTN provides redundancy of reference stations in the solution, such that if observations from one reference station are not available, a solution is still possible since the observations are gathered and processed in a common network adjustment. GNSS-RTN provides multiple benefits over traditional RTK, such as eliminating the need to establish a base station, integrity self-checking by the RTN, and a common reference coordinate system. Limitations include the high cost to establish and maintain such a system and accuracy being limited by the quality of the cellular phone connection [17]. Richter provided a brief comparison of RTK, GNSS-RTN, and PPP. PPP allows for much more extensive coverage with precision to the centimeter level, but the time to access the data (initialization time) is not favorable. GNSS-RTN, on the other hand, eliminates the need for individual base stations and allows for more extensive coverage, but users are still limited to the extent of the network [18]. The characteristics of the aforementioned positioning methods are presented in Table 1.

**Table 1.** Characteristics of Various GNSS Positioning Methods.

| Attributes | GPS | RTK | PPP | RTK-PPP | GNSS-RTN |
|---|---|---|---|---|---|
| Accuracy | 1–10 m | ~1 cm | 3–10 cm | 2–8 cm | 1–5 cm |
| Initialization time | Immediate | Immediate | Slow (~20 min) | Fast (<1 min) | Immediate |
| Coverage area | Global | Local | Global | Regional | Network |
| Bandwidth requirements | Low | High | Low | Moderate | High |
| Infrastructure cost | N/A | Low | N/A | Moderate | High |

Adapted from GPS World Magazine [19].

## 3. Geospatial Data Use in Transportation Applications

Since the GPS became open to civilian applications around three decades ago, geospatial data has increasingly found applications in the transportation industry. This section aims to provide a brief overview of the use of GPS technology in some of the major established transportation applications.

### 3.1. Vehicle Navigation Systems

The availability of accurate location data offers improved efficiencies and safety for all surface transportation systems. The GNSS (commonly known as GPS) navigation systems have been used as built-in devices in the newer models of vehicles, as standalone navigation devices, and in most smartphone devices as built-in applications. However, the GPS location service has a positioning error up to several meters. GPS data have been used in land-vehicle-navigation (LVN) and fused with Inertial Navigation System (INS) data to obtain a relatively accurate position of the vehicle [20]. The inertial navigation system (INS) is a self-contained navigation tool that tracks the position and orientation of an object relative to a known starting point based on measurements provided by accelerometers and gyroscopes. However, these technologies still rely on GPS signals and only work well in areas where GPS reception is not weak or compromised by substantial multipath issues. In GPS-challenged environments, such as urban canyons and forested streets, or even driving in congested traffic adjacent to several vehicles, the GPS-INS technologies may not function well. Toledo-Moreo et al. investigated challenges for navigation systems such as lane-level positioning, map matching, and the quality of the navigation system. Measurements from a GNSS receiver, an odometer, a gyroscope, and enhanced digital maps are all combined in the proposed system. The proposed system showed good results in terms of positioning, map matching, and integrity [21].

### 3.2. Vehicle Fleet Management

GNSS applications are also used in fleet management services (FMSs) [22,23]. Specifically, GNSS constitutes the backbone of the Automatic Vehicle Location (AVL) technology used in fleet management services. Fleets of trucks, couriers, taxis, and other commercial vehicles can be managed using a positioning system and two-way communications between vehicles and a central control center. This way, companies can improve scheduling, reduce operating costs, and enhance effective distribution, thus saving energy [23–25]. Besides, traffic violations such as abnormal driving behavior, speeding, driving illegal routes, etc., can be reduced with the successful implementation of GNSS-based FMSs [23]. Moreover, with the use of geospatial-based FMS along with an incident management system (IMS), emergency vehicles can be dispatched and monitored. The GNSS and its associated FMSs have also found widespread use in operating mass transit systems, road construction and maintenance crews, and emergency vehicles. Most public transportation agencies have implemented AVL systems within their fleets to monitor the schedules and operations on available routes. A GNSS-enabled bus routing software provides the utility of better maintaining the buses on time. It also provides the most up-to-date information to riders about ongoing trips on buses or metro trains, such as accurate arrival times at different stops on the route. The public transit operators can receive alerts about potential issues impacting routes, delivery, and fleet workflows.

Another interesting application of FMSs is locating and tracking snowplows during snow emergencies. To ensure that snow removal is efficient and that plows and salt trucks are dispatched in a timely fashion, these vehicles are equipped with AVL technology and two-way communication with a central facility. Routes that have been plowed are mapped and updated to show the latest information. With the availability of GNSS-RTN accurate position services in real-time, it would be even feasible to identify which lane of a multilane highway has been plowed. The Alaska Department of Transportation uses positioning data from real-time kinematics (RTK) for a special fleet of snowplows known as smart snowplows. These plows are outfitted with GNSS receivers and receive corrections from an

RTK base station. Combined with collision avoidance technologies and GIS, snowplows can alert the driver visually and haptically through vibrations in the seat when the snowplow is drifting outside of a lane [26]. While snowplows are not able to operate at high speeds, the technology allows for safe operation in conditions where visibility is limited [27].

### 3.3. Transportation System Management

Traffic congestion is a serious problem, especially on urban freeways. The most important factors used in Integrated Traffic Management Systems (ITMS) for detecting, monitoring, and controlling traffic congestion are travel time, speed, and delay. The real-time GNSS data can be used to determine travel time, speed, and delays and thus can be utilized for ITMS. The main advantage of monitoring congestion using GNSS data is that real-time information on travel time and speeds can be obtained in an accurate, economical, and timely manner [28,29]. This system can be utilized for daily congestion management in real-time or for annual congestion monitoring based on periodic measurements during a particular season or infrequently throughout the year. In addition to monitoring traffic operations during congestion, the system can also precisely locate incidents that occur on the freeway [28].

Geospatial techniques and data are also used in transportation asset management as they can be referenced easily to locations in pre-existing databases. Some highway agencies utilize GNSS data along with various imaging technologies to determine pavement condition down to as fine a scale as needed [30–32]. In addition, to establish an inventory of roadway networks along with attributes, LiDAR, along with integrated GNSS and INS, is used on a mobile platform or a vehicle, known as a mobile LiDAR system, to capture roadway markings, assets, and cross-sections [33]. The same technology has been used to collect locations for pedestrian infrastructure [31], guardrails, culverts, and bridges [34]. The Ohio Department of Transportation (DOT) divided its road network into 1.9 million 0.01-mile segments to integrate data for legacy systems, newer web applications, and GPS [34]. In addition, GNSS technology, along with other sensors such as LiDAR, vehicle vibrations, and digital cameras, are utilized in the automated survey of pavement surface conditions [35–37]. In this regard, commercially available systems offer solutions integrating GNSS and imaging systems that can be mounted onto a cargo van for pavement evaluation [38,39].

### 3.4. Highway Construction

The GNSS technology is widely utilized in the construction industry for infrastructure projects and mapping applications. With the availability of differential GPS (DGPS), it is now possible to survey a large area quickly and efficiently to create a digital map of a highway network [40].

An Automated Machine Guidance (AMG) system relies on GNSS technology to guide or control heavy construction equipment, including dozers, motor graders, excavators, pavers, and other equipment. In the United States, the Oregon and Iowa Departments of Transportation are considered pioneers in the implementation of AMG, with both agencies starting using the AMG technology over 15 years ago. Some of the benefits of an AMG system are better control of quantities, increased productivity, increased accuracy and precision, more uniform surfaces, fuel savings, and optimized efficiency in surveying [41,42]. Similarly, uniform asphalt density is an important characteristic of pavement performance. Consequently, intelligent compaction (IC) aims to solve this problem by outfitting compactors with sensors such as accelerometers, temperature sensors, GPS receivers, and onboard computers to calculate and display asphalt density in real-time. The U.S. Federal Highway Administration (FHWA) has found that IC is effective in not only reaching the target density but also achieving a more uniform density [43,44]. Intelligent compaction can also be used to determine compaction curves, which could make the compaction operation more efficient by potentially calling for fewer passes. The GNSS data are used for tracking

the compactor as it moves along the project and for mapping the results onto the onboard computer for easy viewing and post-processing.

Furthermore, many construction equipment manufacturers utilize GNSS technology in their dozers for more accurate and efficient operations. Specifically, the GNSS is used on the dozers to adjust the blade tilt and lift as the dozer moves across the project site [45–47]. Using a dual-GNSS solution for blade tilt and lift can offer millimeter accuracy on finished grades and improve accuracy while reducing costs [47].

### 3.5. Aviation and Marine Transportation

The GNSS has extensive applications in air and water transportation systems. In air transportation, GNSS technology has enabled the creation of waypoints without needing to establish a physical infrastructure, thus resulting in increased efficiency and safety [48]. The aviation sector obtains significant economic and environmental benefits from the GNSS technology. It enables aircraft to fly a direct path from origin to destination using the most fuel-efficient routes and navigate complicated terrain at low altitudes to minimize fuel consumption, noise, and carbon emissions [49]. The GNSS is also used to guide aircraft while approaching airports and offers continuous support in smooth descent operations. While there is very little chance of faults or failure in the GNSS technology onboard, the civil aviation community has supplemented the real-time GPS data with a ground-based augmentation system (consisting of three or four reference receivers located on the airport property) that detects and removes errors due to these faults and failure [49].

Similarly, in the ocean, it is near impossible to establish physical wayfinding infrastructure due to the depth of the water. GNSS has drastically changed maritime navigation, whether it is the largest tankers and container ships, which often have three or more GNSS receivers for redundancy, or small leisure ships using inexpensive GNSS handsets. The GNSS provides marine operations a safe and effective way to determine the location, speed, and heading of a vessel. This allows vessels to operate efficiently and safely, especially in tight areas such as harbors [50]. Moreover, GNSS technology, along with geographic information system (GIS) software, is utilized to facilitate the automation of the pick-up, transfer, and placement process of containers in the world's largest port facilities and consequently make the operations and management of containers efficient at port facilities.

### 3.6. Unmanned Aerial Vehicles (UAVs)

Unmanned aerial vehicles (UAVs), commonly known as drones, are smaller aircraft without a pilot with many potential and practical applications both within and outside the transportation field. Specifically, UAVs have increasingly been used in reconnaissance and surveillance (both for civilian and military purposes), surveying and mapping (e.g., photogrammetry and aerial mapping), exploration, and even recreation. UAVs equipped with radars and/or cameras have been utilized for fast and cost-effective data collection in the sphere of traffic operations, highway infrastructure management, forestry, disaster management, agriculture, humanitarian activities, and other applications. In most of the aforementioned applications, the GNSS is the primary source of navigation for UAVs operating over large areas, and it is crucial to determine the position, altitude, and speed of the aircraft. The use of GNSS technology in UAVs is the key to their accuracy and operational safety [51–53]. Whether the UAV is operated by a ground operator or flies fully autonomously in accordance with a pre-programmed flight plan, GNSS navigation techniques offer consistent accuracy, provided that the UAV receives sufficient satellite signals throughout the flight. For more accurate and reliable navigation, some UAVs have been using integrated navigation systems combining inertial sensors and GNSS data.

Recently UAVs have been experimented with for delivering items, for instance, the Amazon Prime Air service parcel delivery to customers and delivery of important medical aids for patients in emergency situations [54–56]. Findings from a recent study show that the average positional error of landing position between the actual landing spot and the desired landing spot in the UAV delivery experiment was in the order of 1.1 m using

the GNSS technology [55]. Obviously, the use of GNSS-RTN technology to support UAV navigation is expected to alleviate the errors in the landing position.

It is important to know the exact location of the UAV at all times of its operation for safety purposes, especially in areas with heavy air traffic, in order to avoid collisions. In addition, GNSS technology is used to prevent the UAV from flying into restricted areas such as airports or any other restricted airspace.

## 4. GNSS-RTN: Emerging Transportation Applications

The previous sections discussed the GNSS-RTN technology and some of the current transportation applications where GPS data is used, including the highly accurate geospatial data. This section will focus on the emerging applications of the highly accurate location data, specifically in transportation, offered by the GNSS-RTN technology. The current evolution of automated transportation systems, which constitute the most important development since the advent of the automobile, is one good example of the role this technology could play in future transportation systems.

### 4.1. Connected and Automated Vehicles (CAVs)

In recent years, auto manufacturers and technology companies have been working tirelessly to develop autonomous vehicles (AVs) that would revolutionize the way people and goods are moved on the transportation network. Thus far, the premise of automated vehicles is to use advanced sensors such as LiDAR, radar, and cameras (among other sensors) in gathering information on the surrounding environment for vehicle control. This approach is reasonable in urban environments and particularly on higher-class highways and streets where the road is paved and delineated with appropriate signage and markings. However, in rural areas, many highways are built to lower standards, lack appropriate signage and markings, and may be unpaved. This presents a difficult challenge for the use of AVs, which are currently being developed and experimented on, in rural environments. Therefore, the use of high-precision GNSS-RTN location data provides a promising approach to addressing complex rural environments when used with precise digital maps of rural roadways and infrastructure. Currently, digital maps have become commonplace worldwide for many advanced applications, especially in urban environments. Digital maps for rural roadways and infrastructure, though not as well established in practice, can be developed to support advanced applications in rural areas. Currently, LiDAR, cameras and other sensors in autonomous vehicles are prone to failure when used in suboptimal weather conditions such as snow, rain, and fog. Such conditions have no impact on GNSS-RTN technology, making it a highly reliable complement to LiDAR and other sensors on AVs, therefore improving safety [57]. High-level precision in positioning is necessary to guarantee the safety of AV users. The location precision provided by GNSS-RTN technology is high, but its availability was scarce when the technology was first introduced due to the few navigation satellites in orbit. The higher number of satellites in orbit allows for more reliable use of GNSS technology by improving error correction. It has been found that GNSS technology, as it is today, is accurate enough to provide, with high confidence, lane determination—one of the biggest concerns with automated vehicles [57]. Besides, GNSS-RTN technology would enable connected and automated vehicles (CAVs) to send (broadcast) and receive accurate location information of traffic conditions, breakdown incident alerts, road pavement conditions sensed by sensors, crash positions, and specific road restrictions.

In a white paper for Swift Navigation, Joubert et al. discussed how advancements in GNSS and corrections networks such as GNSS-RTNs will enable more advanced localization in autonomous vehicles. Localization is the process of the vehicle setting itself into its surroundings. This is done with many data sources such as radar, LiDAR, cameras, GNSS, and GNSS corrections networks, such as GNSS-RTN [57]. The most recent studies highlighted in the white paper demonstrated a 95th percentile horizontal accuracy of 0.14 m in an urban environment using GNSS-RTN. A fixed solution was available for 87 percent

of the study interval. This comes close to the stated required accuracy of 0.1 m in urban environments [58,59].

### 4.2. Applications in Smart Cities

Another important emerging application of the GNSS-RTN technology is expected to support the smart cities concept in terms of urban mobility and safety by providing precise location data in real-time. In a smart city context, various elements of infrastructure are tagged with their spatial coordinates, and the end-users require accurate location and timing information. To this end, the GNSS-RTN accurate geospatial data is expected to satisfy these needs and would support the monitoring system in observing the assets within the required level of accuracy. The easy and accurate locating of assets is essential to maintaining and operating the assets and all physical/spatial features. The GNSS-RTN can be incorporated into intelligent transportation systems (ITS), parking management, access control, and facilitate the path for smart cities. A proposed location-based service, Galileo EnHancement as BoOste of the Smart CiTies (GHOST), uses the European GNSS high-precision positioning to monitor road deterioration, police irregular parking, and report street lighting anomalies. The GHOST concept promises to increase the performance and efficiency of a city's infrastructure while reducing its monitoring costs [60]. Similarly, GNSS technology and Wi-Fi signals were used to develop a platform designed to test and assess the positioning of navigation technologies with the European project HANSEL. The platform processes GNSS snapshots transmitted from smartphones. The processed data is then shared through Wi-Fi access point location services to all smart city users [61].

### 4.3. Transportation Infrastructure

The level of accuracy of the GNSS-RTN location data and the lack of latency would open the door for other automated processes in preserving and managing the transportation system infrastructure. One such application is lane striping operations and rumble strip installation, where location data could be used to automate the process, thus increasing the efficiency and reducing the costs involved in the traditional human-controlled processes.

### 4.4. Highway Work Zones

Safety at road construction sites is a serious concern for most transportation agencies. A construction site is a complex system consisting of workers, machines, materials, activities, and facilities. In recent years, leveraging the rapid advancement in technology, the construction industry has put forward an innovative management model known as smart construction sites, which shifts the construction industry from labor-intensive ways to automation and data-driven ones. At the construction site, tracking the construction equipment and machinery is central for the monitoring of safety, productivity, and sustainability-related practices [62]. The GNSS-RTN is one of the technologies deployed at smart construction sites and is used to track the operations of heavy machinery [63,64]. The high accuracy in positioning services using GNSS-RTN would make the construction industry safer while enhancing productivity and efficiency.

### 4.5. Advanced Vehicle Control Systems

To enhance safety and comfort in highway traffic, GNSS-based applications have gained increased use in modern-day advanced driver assistance systems (ADAS). Whereas the traditional driver assistance systems such as the antilock braking system (ABS) make use of only onboard sensors, recent and future ADAS applications would account for the whole road environment that would be supported by an onboard GNSS receiver. For instance, the efficient and active detection of accurate location of stationary objects and moving vehicles in the surroundings is essential for safe vehicle control. Furthermore, the potential application of the GNSS-RTN would enhance the adaptive cruise control (ACC) systems for vehicles in situations where the vehicles cannot locate each other using onboard sensors, such as traversing a sharp curve and driving on hilly terrain. For instance, the

ACC system currently in use effectively keeps a safe distance from the leading vehicle; however, the poor functioning of sensors in inclement weather results in restricted and limited ACC system operations. In such a scenario, the functionalities of ACC systems can be significantly improved by incorporating GNSS-RTN technology along with vehicle-to-vehicle (V2V) communication, which would enable long-range detection of the surrounding road environment [65].

Additionally, the ubiquity and reliability of highly accurate location data offered by GNSS-RTN technology would increase the effectiveness of other in-vehicle advanced systems, such as advanced vehicle control and collision avoidance systems. These systems require lane-level (sub-meter) positioning accuracy for a vehicle to distinguish its lane from the adjacent lane.

## 5. Concluding Remarks

This article provided an overview of the GNSS-RTN technology and the use of geospatial data in some of the established transportation applications, as well as a perspective on some of the emerging and future applications of the GNSS-RTN technology in the transportation field.

From unmanned aerial vehicles (UAVs) to AVs, all modern-day transportation relies on a steady stream of signals and information from external sources for localization, route planning, perception, and general situational awareness. This includes reliance on positioning, navigation, and timing information. A highly accurate positioning service is essential both for short-range driving control and long-range navigation and planning applications. The role of the GNSS-RTN system will only increase in the future, especially with the advent of automated transportation systems. One challenge in reaping the full benefits of GNSS-RTN technology is the geographic coverage of the network (s), especially for applications that require highly accurate geospatial data. For example, in the United States, more than half of the states have already established statewide GNSS-RTN systems; however, many other states do not have systems in place. This can be a serious issue in using this technology for autonomous vehicle control at the national level, as this requires seamless access to accurate geospatial data in real-time over the whole geographic area. Similar situations may exist in other countries and geographic regions. Governments could play an important role in expediting the full implementation of this technology over the whole geographic extent of a country or region.

As confirmed by recent studies, the benefits of GNSS-RTN within existing applications far outweigh the implementation costs of such systems, not to mention the potential of technology in supporting many future transportation applications. The GNSS-RTN has the potential, through providing accurate geospatial data in real-time, to enhance vehicle safety and operations, automate road construction and maintenance, and examine the condition of pavement surfaces for pavement management and rehabilitation.

This article provided an overview of GNSS positioning technologies with special emphasis on the GNSS-RTN and its applications in current and emerging transportation applications. Such information is deemed valuable and important to transportation professionals working in areas where geospatial location data is required.

The authors recommend further investigation into the geospatial data requirements for some of the emerging advanced transportation applications, especially in regard to accuracy and sampling rate, and the potential role new communications technologies (e.g., 5G network) could play in meeting these requirements. As uncertainty has importance in the performance of transportation systems [66], it will be an interesting research direction to incorporate uncertainties in future research.

**Author Contributions:** Conceptualization, A.A.-K. and S.R.; literature review, S.R., R.T. and B.M.; writing—original draft preparation, S.R., R.T. and B.M.; writing—review and editing, A.A.-K. and S.R.; supervision, A.A.-K.; project administration, A.A.-K.; funding acquisition, A.A.-K. All authors have read and agreed to the published version of the manuscript.

**Funding:** This research was funded by the Western Transportation Institute's Small Urban, Rural, and Tribal Center on Mobility (SURTCOM, grant # 4W8963) and the Montana Department of Transportation (MDT, grant # 4W8950).

**Institutional Review Board Statement:** Not applicable.

**Informed Consent Statement:** Not applicable.

**Data Availability Statement:** Not applicable.

**Conflicts of Interest:** The authors declare no conflict of interest.

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
