# Peer review of "The Role of GNSS-RTN in Transportation Applications"

_encyclopedia, doi:10.3390/encyclopedia2030083_

Round 1
Reviewer 1 Report
General Comment
The manuscript is devoted to the very actual and modern problem of GNSS-RTN implementation for the transportation applications. Unfortunately the text is written in very general manner and do not consists of really novel and interesting details. Instead of GNSS-RTK technology solutions and its shortcomings and advantages the manuscript describes the list of the technology applications. However such a list is well known and obvious and it informs nothing new for the readers. The main problems of the GNSS-RTN technologies implementations of transportation stays out of the authors’ attention. I would recommend to the authors address first in such a problem as: 1) multipath and sudden PDOP degradation at the moving objects; 2) intersystem biases while multi-GNSS is in usage; 3) CORS network parameters, characteristics and limitations for its usage inside GNSS-RTN technologies for transportation applications. Based on the said I would recommend major revision and re-submission. Hope that my remarks below will help the authors to improve their paper.
Major Comments
- According to the title, the manuscript is devoted to the problem of GNSS-RTN implementation in transportation applications. On the other hand, the authors spend time and manuscript space to mix this problem with another problem of the structural health monitoring. In my opinion, the last problem is more in proxy of the surveying then transportation. GNSS solutions for surveying differ from GNSS solutions for transportations significantly and have much higher RNP (required navigation parameters). Due to the limitation of the manuscript and for better understanding of the main issue of the manuscript I would recommend to the authors exclude all the information relating to the surveying problems and focus on the very GNSS solutions for transportation applications.
- Section 3: In a present form, it mostly contains general outlook and references. However It would be very useful and interesting for readers to see concrete values of positioning accuracy, integrity and RNP availability provided by traditional positioning services (two-way radio links, GSM-based triangulation et al.), standalone GPS, integrated GPS-INS-Sensors, traditional RTK. It is necessary to compare GNSS-RTN and the above mentioned positioning and geo-spatial services to see all the advantages of GPS-RTN for the transportation applications clearer.
- Lines 316-317: “…the use of high precision GNSS-RTN location data provides a promising approach to addressing complex rural environments…”. Yes, I agree, but it is only possible based on the precise digital maps of the transportation routes. There is no big sense in precise positioning when there is nothing to compare and check a route. Unfortunately, I have not seen even a word about this principal matter providing all GNSS-RTN advantages for transportation applications. I would recommend to the authors including this principal issue in the manuscript.
- Line 421: “This article provided an overview of the GNSS-RTN technology…”. In my opinion, I would not said that the manuscript contains “overview of GNSS-RTN technology”. Instead of it, I see overview of the applications mixing several principally different areas (not only the transportation but also structural health control, multi-sensors solutions et al.). As I told earlier (please see comment#1), it would be better if the authors had focused on the very transportation problems involving GNSS-RTN technologies (as it is been declared in the headline). It would be more than enough for the one manuscript.
Also I would recommend to the author addressing to the papers below as good sources to take in account in this review paper (I think it would be useful for the readers to find these papers in the paper reference list for self-reading):
· Rizos C. Trends in GPS Technology and Applications. Available at https://www.researchgate.net/publication/267254924_Trends_in_GPS_Technology_Applications
· Demyanov V.V., Imarova O.B., Garmysheva E.S. and Yasyukevich Yu.V. Main current problems and directions for the development of GNSS technologies (2018). In Proc. of the VI International Symp. On Innovation and Sustainability of Modern Railway (ISMR 2018). China Railway Publishing House. Beijing. 2018. Available at:https://www.researchgate.net/publication/328015392_Main_problems_of_GNSS_technologies
Minor comments
- Fig 1 seems unclear and it is not described enough: a) please make all the inscriptions bigger, nothing seen there; b) please mark the data processing center; c) what is virtual reference station and why NMEA stream uses instead of ranging and correction data?
- Lines 85-86: “….providers of GNSS-based services and products around the globe…”. In my opinion, it would be very useful for the review paper to provide not only a picture with the GPS-RTN service coverage in the US but also in the world. I would recommend to the authors expand the Fig 2 correspondently.
- Line 419: Please add dot after the words “…the adjacent lane”.
Author Response
Dear Reviewer,
Please see the attached response to your comments.

Reviewer 2 Report
This article provided an overview of the GNSS-RTN technology and the use of geo-spatial data in some of the established transportation applications as well as a perspective on some of the emerging and future applications of the GNSS-RTN technology in the transportation field. From unmanned aerial vehicles (UAVs) to AVs, all modern-day transportation relies on a steady stream of signals and information from external sources for localization, route planning, perception, and general situational awareness. This includes reliance on positioning, navigation, and timing information. A highly accurate positioning service is essential both for short-range driving control and long-range navigation and planning applications. The role of the GNSS-RTN system will only be increasing in the future, especially with the advent of automated transportation systems.
The academic contributon of the paper relative to the existing literature should be highlighted?
Who are potential users of the research (i.e., what will benefit from the research), industry, government, or academia?
"The GNSS-RTN has the potential, through providing accurate geospatial data in real-time, to enhance vehicle safety and operations, automate road construction and maintenance, and examine the pavement surface condition for pavement management and rehabilitation." -> "The GNSS-RTN has the potential, through providing accurate geospatial data in real-time, to enhance vehicle safety and operations, automate road construction and maintenance, and examine the pavement surface condition for pavement management and rehabilitation. Uncertainties are not considered in this study (Wang and Wu, 2021) and it will be an interesting research direction to incorporate uncertainties in the research."
Wang, W. and Wu, Y. 2021. Is uncertainty always bad for the performance of transportation systems? Communications in Transportation Research, 1, 100021.
Author Response

(The authors gave the same response as above.)

Reviewer 3 Report
1. In the introduction purpose of this article should be clearly stated.
2. In the article it is recommended to emphases / highlight the critical insights of the authors themselves
3. The conclusions should indicate the further direction of the research.
Author Response

(The authors gave the same response as above.)

Round 2
Reviewer 1 Report
No additional comments
Reviewer 2 Report
This version is acceptable.
Reviewer 3 Report
Good luck